# PointTAD: Multi-Label Temporal Action Detection with Learnable Query Points

**Jing Tan** [1]*  **Xiaotong Zhao** [2]  **Xintian Shi** [2]  **Bin Kang** [2]  **Limin Wang** [1,3]†

[1]State Key Laboratory for Novel Software Technology, Nanjing University
[2]Platform and Content Group (PCG), Tencent    [3]Shanghai AI Lab

`jtan@smail.nju.edu.cn, {davidxtzhao,tinaxtshi,binkang}@tencent.com, lmwang@nju.edu.cn`

## Abstract

Traditional temporal action detection (TAD) usually handles untrimmed videos with small number of action instances from a single label (e.g., ActivityNet, THU-MOS). However, this setting might be unrealistic as different classes of actions often co-occur in practice. In this paper, we focus on the task of multi-label temporal action detection that aims to localize all action instances from a multi-label untrimmed video. Multi-label TAD is more challenging as it requires for fine-grained class discrimination within a single video and precise localization of the co-occurring instances. To mitigate this issue, we extend the sparse query-based detection paradigm from the traditional TAD and propose the multi-label TAD framework of PointTAD. Specifically, our PointTAD introduces a small set of learnable query points to represent the important frames of each action instance. This point-based representation provides a flexible mechanism to localize the discriminative frames at boundaries and as well the important frames inside the action. Moreover, we perform the action decoding process with the Multi-level Interactive Module to capture both point-level and instance-level action semantics. Finally, our PointTAD employs an end-to-end trainable framework simply based on RGB input for easy deployment. We evaluate our proposed method on two popular benchmarks and introduce the new metric of detection-mAP for multi-label TAD. Our model outperforms all previous methods by a large margin under the detection-mAP metric, and also achieves promising results under the segmentation-mAP metric. Code is available at https://github.com/MCG-NJU/PointTAD.

## 1  Introduction

With the increasing amount of video resources on the Internet, video understanding is becoming one of the most important topics in computer vision. Temporal action detection (TAD) [50, 23, 21, 4, 10, 19, 36] has been formally studied on traditional benchmarks such as THUMOS [15], ActivityNet [14], and HACS [48]. However, the task seems impractical because their videos almost contain non-overlapping actions from a single category: $85\%$ videos in THUMOS are annotated with single action category. As a result, most TAD methods [23, 21, 43, 5, 34] simply cast this TAD problem into sub-problems of action proposal generation and global video classification [40]. In this paper, we shift our playground to the more complex setup of multi-label temporal action detection, which aims to detect all action instances from multi-labeled untrimmed videos. Existing works [9, 16, 38, 8] in this field formulate the problem as a dense prediction task and perform multi-label classification in a frame-wise manner. Consequently, these methods are weak in localization and fail to provide the instance-level detection results (i.e., the starting time and ending time of each instance). In analogy to image instance segmentation [22], we argue that it is necessary to redefine multi-label TAD as a

---

*Work is done during internship at Tencent PCG.  †Corresponding author.

36th Conference on Neural Information Processing Systems (NeurIPS 2022).

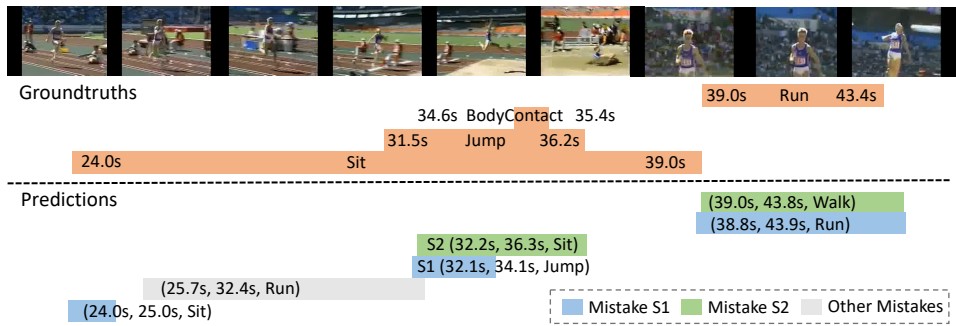

Figure 1: Illustration of action predictions by segment-based action detectors in multi-label TAD.

instance-level detection problem rather than a frame-wise segmentation task. In this sense, multi-label TAD results not only provide the action labels, but also the exact temporal extent of each instance.

Direct adaptation of action detectors is insufficient to deal with the challenges of concurrent instances and complex action relations in multi-label TAD. The convention of extracting action features from **action segments** [4, 31, 47, 41] lacks the flexibility of handling both important semantic frames inside the instance as well as discriminative boundary frames. Consider the groundtruth action of *"Jump"* in Fig. 1, segment-based action detectors mainly produce two kinds of error predictions, as in type $S_1$ and type $S_2$. $S_1$ successfully predicts the correct action category with an **incomplete** segment of action highlights, whereas $S_2$ does a better job in locating the boundaries yet get **misclassified** as *"Sit"* due to the inclusion of confusing frames. In addition, most action detectors [47, 31, 25] are inadequate in processing sampled frames and classifying fine-grained action classes. They often exploit temporal context modeling at a single level and ignore the exploration of channel semantics.

To address the above issues, we present PointTAD, a sparse query-based action detector that leverages learnable query points to flexibly attend important frames for action instances. Inspired by RepPoints [45], the query points are directly supervised by regression loss. Given specific regression targets, the query points learn to locate at discriminative frames at action boundaries as well as semantic key frames within actions. Hence, concurrent actions of different categories can yield distinctive features through the specific arrangement of query points. Moreover, we improve the action localization and semantics decoding by proposing the Multi-level Interactive Module with dynamic kernels based on query vector. Direct interpolation or pooling at each query point lacks temporal reasoning over consecutive frames. Following deformable DETR [53], we extract point-level features with deformable convolution [6, 52] from a local snippet to capture the temporal cues of action change or important movement. At instance-level, both temporal context and channel semantics are captured with frame-wise and channel-wise dynamic mixing [35, 11] to further decode the distinctive features of simultaneous actions.

PointTAD streamlines end-to-end TAD with joint optimization of the backbone network and action decoder without any post-processing technique. We validate our model on two challenging multi-label TAD benchmarks. Our model achieves the state-of-the-art detection-mAP performance and competitive segmentation-mAP performance to previous methods with RGB input.

## 2 Related Work

**Multi-label temporal action detection.** Multi-label temporal action detection has been studied as a multi-label frame-wise classification problem in the previous literature. Early methods [29, 30] paid a lot of attention on modeling the temporal relations between frames with the help of Gaussian filters in temporal dimension. Other works integrated features at different temporal scales with dilated temporal kernels [9] or iterative convolution-attention pairs [8]. Recently, attention has shifted beyond temporal modeling. Coarse-Fine [16] handled different temporal resolutions in the slow-fast fashion and performed spatial-temporal attention during fusion. MLAD [38] used multi-head self-attention blocks at both spatial and class dimension to model class relations at each timestamp. In our proposed method, we view the task as a instance-level detection problem and employ query-based framework with sparse temporal points for accurate action detection. In addition, we study the temporal context at different semantic levels, including inter-proposal, intra-proposal and point-level of modeling.

**Segment-based representation.** Following the prevailing practice of bounding boxes [20, 13, 32, 1] in object detection, existing temporal action detectors incorporated action segments heavily with three kinds of usage: as anchors, as intermediate proposals, and as final predictions. Segments as anchors are explored mainly in anchor-based frameworks. These methods [28, 26, 47, 31] used sliding windows or pre-computed proposals as anchors. Most TAD methods [47, 19, 43, 4, 50, 49] use segments as intermediate proposals. Uniform sampling or pooling are commonly used to extract features from these segments. P-GCN [47] applied max-pooling within local segments for proposal features. G-TAD [43] uniformly divided segments into bins and average-pooled each bin to obtain proposal features. AFSD [19] proposed boundary pooling in boundary region to refine action feature. Segments as final predictions are employed among all TAD frameworks, because segments generally facilitate the computation of action overlaps and loss functions. Instead, in this paper, we do not need segments as anchors and directly employ learnable query points as intermediate proposals with iterative refinement. The learnable query points represent the important frames within action and action feature is extracted only from these keyframes rather than using RoI pooling.

**Point-based representation.** Several existing works have used point representations to describe keyframes [12, 37], objects [45, 11], tracks [51], and actions [18]. [12, 37] tackled keyframe selection by operating greedy algorithm on spatial SIFT keypoints [12] or clustering on local extremes of image color/intensity [37]. These methods followed a bottom-up strategy to choose keyframes based on local cues. In contrast, PointTAD represents action as a set of temporal points (keyframes). We follow RepPoints [45] to handle the important frames of actions with point representations and refine these points by action feature iteratively. Our method directly regresses keyframes from query vectors in a top-down manner for more flexible temporal action detection. Note that PointTAD tackles different tasks from RepPoints [45]. We also built PointTAD upon a query-based detector, where a small set of action queries is employed to sparsely attend the frame sequence for potential actions, resulting in an efficient detection framework.

**Temporal context in videos.** Context aggregation at different levels of semantics is crucial for temporal action modeling [39] and has been discussed in previous TAD methods. G-TAD [43] treated each snippet input as graph node and applied graph convolution networks to enhance snippet-level features with global context. ContextLoc [54] handled action semantics in hierarchy: it updated snippet features with global context, obtained proposal features with frame-wise dynamic modeling within each proposal and modeled the inter-proposal relations with GCNs. Although we considered the same levels of semantic modeling, our method is different from ContextLoc. PointTAD focuses on aggregating temporal cues at multiple levels, with deformable convolution at point-level as well as *frame* and *channel* attentions at intra-proposal level. We also apply multi-head self-attention for inter-proposal relation modeling.

## 3   PointTAD

We formulate the task of multi-label temporal action detection (TAD) as a set prediction problem. Formally, given a video clip with $T$ consecutive frames, we predict a set of action instances $\Psi = \{\psi_n = (t_n^s, t_n^e, c_n)\}_{n=1}^{N_q}$, $N_q$ is the number of learnable queries, $t_n^s, t_n^e$ are the starting and ending timestamp of the $n$-th detected instance, $c_n$ is its action category. The groundtruth action set to detect are denoted $\hat{\Psi} = \{\hat{\psi}_n = (\hat{t}_n^s, \hat{t}_n^e, \hat{c}_n)\}_{n=1}^{N_g}$, where $\hat{t}_n^s, \hat{t}_n^e$ are the starting and ending timestamp of the $n$-th action, $\hat{c}_n$ is the groundtruth action category, $N_g$ is the number of groundtruth actions.

The overall architecture of PointTAD is depicted in Fig. 2. PointTAD consists of a **video encoder** and an **action decoder**. The model takes three inputs for each sample: RGB frame sequence of length $T$, a set of learnable query points $\mathbf{P} = \{\mathcal{P}_i\}_{i=1}^{N_q}$, and query vectors $\mathbf{q} \in \mathbb{R}^{N_q \times D}$. Learnable query points explicitly describe the action locations by positioning themselves around action boundaries and semantic key frames, and the query vectors decode action semantics and locations from the sampled features. In the model, the video encoder extracts video features $X \in \mathbb{R}^{T \times D}$ from RGB frames. The action decoder contains $L$ stacked decoder layers and takes query points $\mathbf{P}$, query vectors $\mathbf{q}$ and video features $X$ as input. Each decoder layer contains two parts: 1) the multi-head self-attention block models the pair-wise relationship of query vectors and establishes inter-proposal modeling for action detection; 2) the **Multi-level Interactive Module** models the point-level and instance-level semantics with dynamic weights based on query vector. Overall, the action decoder aggregates the temporal context at **point-level**, **intra-proposal** level and **inter-proposal** level. Finally, we use two

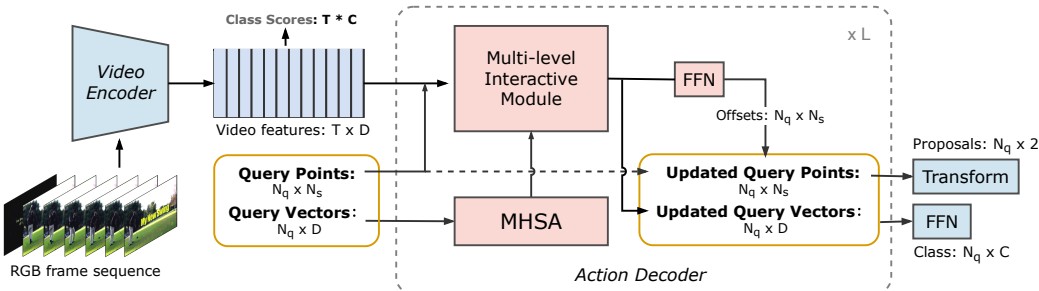

Figure 2: **Pipeline of PointTAD**. It consists of a backbone network that extracts video features from consecutive RGB frames and an action decoder of $L$ layers that directly decodes actions from video features. PointTAD enables end-to-end training of backbone and action decoder without any post-processing of predictions.

linear projection heads to decode action labels from query vectors, and transform query points to detection outputs.

## 3.1 Video Encoder

We use the I3D backbone [3] as the video encoder in our framework. The video encoder is trained end-to-end with the action decoder and optimized by action detection loss to bridge the domain gap between action recognition and detection. For easy deployment of our framework in practice, we avoid the usage of optical flow due to its cumbersome preparation procedure. In order to achieve good performance on par with two-stream features by only using RGB input, we follow [44] to remove the temporal pooling at $\mathrm{Mixed\_5c}$ and fuse the features from $\mathrm{Mixed\_5c}$ with features from $\mathrm{Mixed\_4f}$ as in [24]. As a result, the temporal stride of encoded video features is 4. Spatial average pooling is performed to squeeze the spatiotemporal representations from backbone to temporal features.

## 3.2 Learnable Query Points

Segment-based action representation (i.e., representing each action instance simply with a starting and ending time) is limited in describing its boundary and content at the same time. To increase the representation flexibility, we present a novel point-based representation to automatically learn the positions of action boundary as well as its semantic key frames inside the instance. Specifically, the point-based representation is denoted by $\mathcal{P} = \{t_j\}_{j=1}^{N_s}$ for each query, where $t_j$ is the temporal location of $j^{th}$ query point, and the point quantity per query is $N_s$ and set to 21 empirically. We explain the updating strategy and the learning of query points below.

**Iterative point refinement.** During training, the query points are initially placed at the midpoint of the input video clip. Then, they are refined by query vectors $\mathbf{q}$ through iterations of decoder layers to reach final positions. To be specific, at each decoder layer, the query point offsets are predicted from updated query vector (see Sec. 3.3) by linear projection. We design a self-paced updating strategy with adaptive scaling for each query at each layer to stabilize the training process. At decoder layer $l$, the query points for one query are represented by $\mathcal{P}^l = \{t_j^l\}_{j=1}^{N_s}$. The $N_s$ offsets are denoted $\{\Delta t_j^l\}_{j=1}^{N_s}$. The refinement can be summarized as:

$$\mathcal{P}^{l+1} = \{(t_j^l + \Delta t_j^l \cdot s^l \cdot 0.5)\}_{j=1}^{N_s}, \tag{1}$$

where $s^l = max(\{t_j^l\}) - min(\{t_j^l\})$ is the scaling parameter and describes the span of query points at layer $l$. As a result, the updated step size gets smaller for shorter action, which helps with the localization of short actions. Updated query points from previous layer are inputs to the next layer.

**Learning query points.** The training of query points is directly supervised by regression loss at both intermediate and final stages. We follow [45] to transform query points to pseudo segments for regression loss calculation. The resulted pseudo segments participate in the calculation of L1-loss and tIoU loss with groundtruth action segments in both label assignment and loss computation.

The transformation function is denoted by $\mathcal{T} : \mathcal{P} \rightarrow \mathcal{S} = (t^s, t^e)$. We experiment with two kinds of functions: Min-max $\mathcal{T}_1$ and Partial min-max $\mathcal{T}_2$. *Min-max* is to take the minimum and maximum

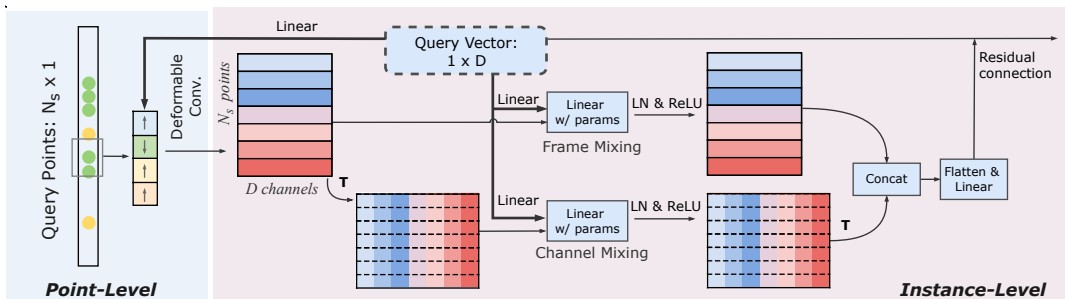

Figure 3: **Multi-level Interactive Module** aggregates action semantics at point-level and instance-level with dynamic parameters.

location from the set of query points as the starting and ending timestamp of the pseudo segment, $\mathcal{T}_1 : \mathcal{P} \to min(\{t_j\}_{j\in\mathcal{P}}), max(\{t_j\}_{j\in\mathcal{P}})$. With Min-max transformation, the query points are strictly bounded within the local segment of the target action instance. *Partial min-max* function is to select a subset of query points $\mathcal{P}_{local}$ and perform the min-max function on them to determine a pseudo segment, $\mathcal{T}_2 : \mathcal{P} \to min(\{t_j\}_{j\in\mathcal{P}_{local}}), max(\{t_j\}_{j\in\mathcal{P}_{local}})$. It allows several query points to aggregate information from outside the action. Empirically, we choose partial min-max by default. For each query, we randomly take $\frac{2}{3}N_s$ points from the query point set to form $\mathcal{P}_{local}$.

### 3.3 Multi-level Interactive Module

Apart from the limitation in the segment-based representation, previous temporal action detectors are also insufficient in decoding the sampled frames. These methods seldom consider semantic aggregation from different levels and at multiple aspects. Accordingly, we present a multi-level interactive module that consider both **local temporal cues at point-level** and **intra-proposal relation modeling at instance-level**, depicted in Fig. 3. To achieve distinct representations for each query, these parameters at both levels are dynamically generated based on the query vector.

**Point-level local deformation.** Paired with the refined point representation, we employ deformable convolutions to extract point features within the local neighborhood. For the $j^{th}$ query point, we predict four temporal offsets $\{o_k\}_{k=1}^4$ from the point location and corresponding weights $\{w_k\}_{k=1}^4$. The query point at frame $t_j$ acts as the center point and is added with temporal offsets to achieve four deformable sub-points. These sub-points characterize the local neighborhood of center points. The features at sub-points are extracted by bilinear interpolation, then multiplied with weights and combined together to get point-level feature $x(j)$. This process can be expressed as:

$$x(j) = \sum_{k=1}^{4} X(t_j + o_k) \cdot w_k. \tag{2}$$

Both the offsets and weights are generated from query vectors $\mathbf{q}$ by linear projection,

$$\mathbf{o} = \text{Linear}(\mathbf{q}) \in \mathbb{R}^{N_q \times 4}, \quad \mathbf{w} = \text{Softmax}(\text{Linear}(\mathbf{q})) \in \mathbb{R}^{N_q \times 4}, \tag{3}$$

where the weights are additionally normalized by softmax for each query.

**Instance-level adaptive mixing.** Faced with the challenge of simultaneous actions, with only temporal modeling, actions with large overlap may result in similar representation and harms the classification. To tackle this problem, we propose adaptive mixing at both frames and channels by using dynamic convolutions. Specifically, the stacked features of query points is denoted by $x \in \mathbb{R}^{N_s \times D}$. Given the query vector $\mathbf{q}$, we generate dynamic parameters for frame mixing and channel mixing:

$$M_f = \text{Linear}(\mathbf{q}) \in \mathbb{R}^{N_s \times N_s}, \tag{4}$$

$$M_{c,1} = \text{Linear}(\mathbf{q}) \in \mathbb{R}^{D \times D'}, \tag{5}$$

$$M_{c,2} = \text{Linear}(\mathbf{q}) \in \mathbb{R}^{D' \times D}. \tag{6}$$

Frame mixing is carried out with dynamic projection followed by LayerNorm and ReLU activation on $N_s$ points to explore intra-proposal frame relations:

$$x_f = \text{ReLU}(\text{LayerNorm}(x^T M_f)) \in \mathbb{R}^{D \times N_s}. \tag{7}$$

Similar to frame mixing, channel mixing uses two bottle-necked layers of dynamic projection on the channel dimension to enhance action semantics:

$$x_c = \text{ReLU}(\text{LayerNorm}(\text{ReLU}(\text{LayerNorm}(x M_{c,1})) M_{c,2})) \in \mathbb{R}^{N_s \times D}. \tag{8}$$

These two mixed features are concatenated along channel and squeezed by linear operations to the size of query vector. The query vector $\mathbf{q}^l$ at the $l^{th}$ layer is then updated with residual connection:

$$\mathbf{q}^{l+1} = \mathbf{q}^l + \text{Linear}(\text{Concat}(x_f^T, x_c)). \tag{9}$$

Finally, the query point offsets and the action labels are decoded from the updated vector by two linear projection heads to produce detection results at each layer.

## 3.4 Training and Inference

**Label assignment.** Similar to all query-based detectors [2, 53, 11, 36], we apply hungarian matcher on (detected) pseudo segments to search for the optimal permutation $\sigma(\cdot)$ for label assignment. The groundtruth set $\hat{\Psi}$ on each video clip is extended with no action $\varnothing$ to the size of $N_q$. The matching cost is formulated as:

$$\mathcal{C} = \sum_{n:\sigma(n) \neq \varnothing} \alpha_{L1} \cdot \mathcal{L}_{L1}(\mathcal{T}(\mathcal{P}_n), \hat{\psi}_{\sigma(n)}) - \alpha_{iou} \cdot \text{tIoU}(\mathcal{T}(\mathcal{P}_n), \hat{\psi}_{\sigma(n)}) - \alpha_{cls} \cdot c_n. \tag{10}$$

By minimizing the matching cost, the bipartite matching algorithm finds the optimal permutation $\sigma_*(\cdot)$ that assigns each prediction with a target. $\alpha_{L1}, \alpha_{iou}, \alpha_{cls}$ is set to $5, 5, 10$ respectively.

**Loss functions.** PointTAD is jointly optimized by localization loss and classification loss. We use $L_1$ loss and iou loss as localization loss:

$$\mathcal{L}_{loc} = \sum_{n:\sigma_*(n) \neq \varnothing} \mathcal{L}_{L_1}(\mathcal{T}(\mathcal{P}_n), \hat{\psi}_{\sigma_*(n)}) + (1 - \text{tIoU}(\mathcal{T}(\mathcal{P}_n), \hat{\psi}_{\sigma_*(n)})). \tag{11}$$

The cross-entropy loss between query labels and target labels is used as classification loss. In addition, to improve the performance under segmentation-mAP, we generate dense classification scores $S \in \mathbb{R}^{T \times C}$ by linear projection from video features $X$. Cross-entropy loss is enforced on $S$ with dense groundtruth $\hat{S} \in \mathbb{R}^{T \times C}$. Therefore, the classification loss is composed of the set prediction loss and the dense per-frame loss:

$$\mathcal{L}_{cls} = \sum_n \mathcal{L}_{ce}(c_n, \hat{c}_{\sigma_*(n)}) + \lambda_{seg} \mathcal{L}_{ce}(S, \hat{S}). \tag{12}$$

The overall loss function is formulated as follows:

$$\mathcal{L} = \lambda_{loc} \cdot \mathcal{L}_{loc} + \lambda_{cls} \cdot \mathcal{L}_{cls}. \tag{13}$$

$\lambda_{cls}, \lambda_{loc}$ and $\lambda_{seg}$ are hyper-parameters that are set to $10, 5, 1$ respectively.

**Inference.** During inference, our PointTAD uses a single linear projection layer followed by LayerNorm and ReLU activation to predict class labels from query vectors. As for localization, we use pseudo segments transformed from query points as the final predictions. These sparse predictions are then evaluated under the detection-mAP metric. Additional dense scores $S$ could be generated at video features for the segmentation-mAP calculation. The sparse predictions are filtered with threshold $\gamma$ and processed with Gaussian kernels to approximate dense scores at each frame. Then, the approximated scores are added with weight $\beta$ to the predicted dense scores for segmentation-mAP calculation. The final dense scores are generated as:

$$S_{final} = \beta \cdot \sum_{n=1}^{N_q} \mathbb{1}_{c_n > \gamma} \cdot \text{Gaussian}(\psi_n) + (1 - \beta) \cdot S. \tag{14}$$

Note that the sparse predictions are adjusted by dense scores only for segmentation-mAP.

Table 1: **Comparison with the state of the art** on the MultiTHUMOS test set and Charades test set, under detection-mAP (%) and segmentation-mAP(%).

| Methods | Modality | MultiTHUMOS | | Charades | |
|---|---|---|---|---|---|
| | | Det-mAP | Seg-mAP | Det-mAP | Seg-mAP |
| R-C3D [42] | RGB | - | - | - | 17.6 |
| Super-event [29] | RGB | - | 36.4 | - | 18.6 |
| TGM [30] | RGB | - | 37.2 | - | 20.6 |
| TGM [30] | RGB+OF | - | 44.3 | - | 21.5 |
| PDAN [9] | RGB | 17.3/17.1$^{\ddagger}$ | 40.2 | 8.5 | 23.7 |
| Coarse-Fine [16] | RGB | - | - | 6.1 | 25.1 |
| MLAD [38] | RGB | 14.2/13.9$^{\ddagger}$ | 42.2 | - | 18.4 |
| MLAD [38] | RGB+OF | - | 51.5 | - | 23.7 |
| CTRN [7] | RGB | - | **44.0** | - | 25.3 |
| CTRN [7] | RGB+OF | - | 51.2 | - | 27.8 |
| AGT [27] | RGB+OF | - | - | - | 28.6 |
| MS-TCT [8] | RGB | 16.2/16.0$^{\ddagger}$ | 43.1 | 7.9 | **25.4** |
| Ours | RGB | 21.5/21.4$^{\ddagger}$ | 39.8 | 11.1 | 21.0 |
| Ours$^{\S}$ | RGB | **23.5/23.4**$^{\ddagger}$ | 41.2 | **12.1** | 22.1 |

$^{\ddagger}$ *indicates detection results excluding NoHuman class.* $^{\S}$*indicates results trained with stronger image augmentation as in [24].*

## 4 Experiments

### 4.1 Datasets and Setup

**Datasets.** We conduct experiments on two popular multi-label action detection benchmarks: MultiTHUMOS [46] and Charades [33]. **MultiTHUMOS** is a densely labeled dataset extended from THUMOS14. It includes 413 sports videos of 65 classes. The average number of distinctive action categories per video in MultiTHUMOS is 10.5, compared with 1.1 in THUMOS14. **Charades** is a large Multi-label TAD dataset that contains 9848 videos of daily in-door activities. The annotations are spread over 157 action classes, with an average of 6.8 instances per video.

**Implementation details.** With I3D backbone network, we extract frames at 10 fps for MultiTHUMOS and 12 fps for Charades. The spatial resolution is set to $192^2$ for both datasets. We report ablations with only `Center_Crop` for training, and report comparison in Tab. 1 with stronger image augmentation following [24]. The video sequence is pre-processed with sliding window mechanism. To accommodate most of the actions, the window size is set to 256 frames for MultiTHUMOS (99.1% actions included), and 400 frames for Charades (97.3% actions included). The overlap ratio is 0.75 at training, and 0 at inference. $N_q$ is set to 48 for both benchmarks. The number of query points per query $N_s$ is 21. The number of deformable sub-points is set to 4 according to the number of sampling points in TadTR [25]. The optimal $\gamma$ is 0.01 for both datasets.

Appropriate **initialization** is required for backbone, query points and point-level deformable convolutions for stable training. Following common practice, the I3D backbone are initialized with Kinetics400 [17] pre-trained weights for MultiTHUMOS and Charades pre-trained weights for Charades. Query points are initialized with constant 0.5 in training and with learned weights in inference. Other possible initializations are explored in ablations. The linear layer to produce deformable offsets are initialized as follows: zeroing for weights and $[1, 2, 3, 4]$ for biases. The weights and biases to generate deformable weights are initialized as zero.

We adopt AdamW as optimizer with 1e-4 weight decay. The network is trained on a server with 8 V100 GPUs. The batch size is 3 per GPU for MultiTHUMOS and 2 per GPU for Charades. The learning rate is set to 2e-4 and drops by half at every 10 epochs. Backbone learning rate is additionally multiplied with 0.1 for stable training.

**Evaluation metrics.** The default evaluation metric for multi-label TAD is segmentation-mAP, which is the frame-wise mAP. In addition, we extend the detection-mAP metric from traditional TAD to further evaluate the completeness of predicted action instances. The detection-mAP is the instance-wise mAP of action predictions under different tIoU thresholds. We report the average mAP as well as mAPs at tIoU threshold set $\{0.2, 0.5, 0.7\}$ for both datasets. The average detection-mAP is

Table 2: **PointTAD Ablation experiments** on MultiTHUMOS. Default setting is colored gray.

(a) **Segments** vs. **Query Points** in query-based action detectors

| | 0.2 | 0.5 | 0.7 | Avg |
|---|---|---|---|---|
| Segment | 33.1 | 20.1 | 9.8 | 19.4 |
| Point | **36.6** | **22.8** | **10.6** | **21.5** |

(b) **Initialization** of query points at training.

| Init. | 0.2 | 0.5 | 0.7 | Avg |
|---|---|---|---|---|
| Uniform(0,1) | 36.9 | 22.6 | 10.0 | 21.3 |
| Norm(0.5,0.3) | **37.0** | 21.9 | 9.4 | 21.2 |
| Const.(0.5) | 36.6 | **22.8** | **10.6** | **21.5** |

(c) **Point2Segment Transformation** $\mathcal{T} : \mathcal{P} \to \mathcal{S}$.

| $\mathcal{T}$ | 0.2 | 0.5 | 0.7 | Avg |
|---|---|---|---|---|
| Min-max | 36.5 | **22.9** | **10.9** | **21.6** |
| Partial Min-max | **36.6** | 22.8 | 10.6 | 21.5 |

(d) **Number of Query Points** $N_s$ per query.

| $N_s$ | 0.2 | 0.5 | 0.7 | Avg |
|---|---|---|---|---|
| 9 | 36.2 | 22.3 | 10.3 | 21.0 |
| 15 | **36.6** | **22.8** | 10.4 | 21.4 |
| 21 | **36.6** | **22.8** | **10.6** | **21.5** |
| 27 | **36.6** | 22.6 | **10.6** | 21.4 |

(e) **Point-Level**: Deformable Convolution.

| Deform. Conv. | 0.2 | 0.5 | 0.7 | Avg |
|---|---|---|---|---|
| ✓ | **36.6** | **22.8** | **10.6** | **21.5** |
| | 35.7 | 22.1 | 9.9 | 20.8 |

(f) **Instance-Level**: different variants of mixing strategy.

| Mixing | 0.2 | 0.5 | 0.7 | Avg |
|---|---|---|---|---|
| Frame Only | 35.7 | 22.3 | 10.2 | 20.9 |
| Channel Only | 34.2 | 21.4 | 9.8 | 20.1 |
| Frame→Channel | 34.3 | 21.4 | 9.2 | 19.9 |
| Channel→Frame | 30.7 | 17.6 | 6.7 | 17.1 |
| Parallel Mixing | **36.6** | **22.8** | **10.6** | **21.5** |

calculated with tIoU thresholds set $[0.1 : 0.1 : 0.9]$. We argue that detection-mAP is more reasonable for an instance detection task.

## 4.2 Comparison with the State-of-the-Art Methods

In Tab. 1, we compare the performance of PointTAD with previous multi-label TAD methods under both segmentation-mAP and detection-mAP. The sparse prediction tuples of PointTAD are converted to dense segmentation scores with Eq. (14) for segmentation-mAP. In order to compare the detection-mAP, we reproduce several previous multi-label temporal action localization methods and convert their dense segmentation results to sparse predictions following [46]. The prediction confidence score of an action with $L_f$ consecutive frames is calculated as:

$$\text{score}(C, p_1...p_{L_f}) = \left(\sum_{i=1}^{L_f} p_i\right) \times \exp\left(\frac{-0.01(L_f - \mu_C)^2}{\sigma_C^2}\right), \tag{15}$$

where $p_i$ is the probability for class $c$ at frame $i$, $\mu_C$ and $\sigma_C$ is the mean and standard derivation of action duration of class $C$ in the training set.

Our PointTAD surpasses all previous multi-label TAD methods by a large margin under detection-mAP, indicating our ability to predict complete actions is beyond previous dense segmentation models. As for segmentation-mAP, we achieve encouraging and comparable results to the previous methods on both benchmarks with a sparse detection framework.

In Fig. 4, we show the qualitative results of PointTAD on the MultiTHUMOS test set compared to the segment-based baseline, PDAN [9] and MS-TCT [8]. PointTAD detects more instances at harder categories, such as "Fall" and "SoccerPenalty". MS-TCT and PDAN perform better at "NoHuman" category. We argue that this is because "NoHuman" class is not a well-defined action category with precise paired action boundaries, whereas PointTAD have to leverage pair relations of boundaries for localization.

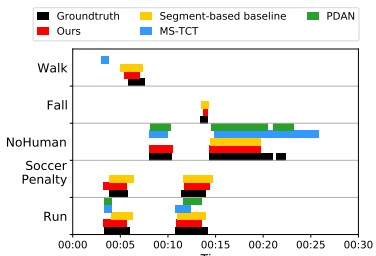

Figure 4: **Qualitative results** on the Multi-THUMOS test set.

## 4.3 Ablation Study

**Segment-based representation vs point-based representation.** In Tab. 2a, we compare the performance between segment-based representation and point-based representation. In the segment-based baseline, actions are represented by segments of paired start-end timestamps. We apply temporal RoI align to the segments as in [41, 49] to retrieve action features. Temporal RoI align divides segments into uniform bins and apply average pooling on each bin to transform frame features for query vector mixing. For fair comparison, the number of bins in RoI align is set to $N_s$ and the parallel dynamic

Table 3: Ablation study w.r.t. **fusion parameter** $\beta$ and **scaling parameter** $s$ on MultiTHUMOS.

(a) **Result fusion parameter** $\beta$: 1 indicates full sparse detection and 0 indicates full dense results. The results are reported under segmentation-mAP.

| $\beta$ | 0 | 0.2 | 0.4 | 0.6 | 0.8 | 0.96 | 1 |
|---|---|---|---|---|---|---|---|
| MultiTHUMOS | 33.0 | **39.8** | 39.2 | 38.1 | 37.3 | 36.8 | 35.9 |
| Charades | | 13.8 | 14.3 | 15.1 | 16.6 | 19.2 | **21.0** | 18.7 |

(b) **Offset scaling parameter** $s$: scale to window size vs. scale to action duration. The results are reported under detection-mAP@tIoU.

| $s$ | 0.2 | 0.5 | 0.7 | Avg |
|---|---|---|---|---|
| scale to clip duration | 36.4 | 21.9 | 9.8 | 21.0 |
| scale to action duration | **36.6** | **22.8** | **10.6** | **21.5** |

mixing is applied to both implementations. Results show that point-based representation significantly outperforms segment-based representation, demonstrating the advantage of adaptive sampling based on temporal points over grid alignment based on segments.

**Study on query point initialization.** We consider three different initialization for query points in training: uniform distribution in $[0, 1]$, normal distribution with mean=0.5 and std = 0.3, and initialization with constant 0.5. Results in Tab. 2b indicates that all three initialization methods are beneficial for the training, with little performance gap in avg-mAP. In addition, normal distribution and uniform distribution achieve higher detection-mAP with lower tIoU threshold, which shows these two initialization methods are weaker at accurate localization under higher tIoU threshold. Empirically, we set constant initialization by default.

**Study on pseudo segment transformation functions.** Tab. 2c shows two alternatives of transformation function $\mathcal{T}$. Min-max and Partial min-max achieve similar performance under average detection-mAP. We choose Partial min-max as the default function because it offers a more relaxed constraint for query points with higher flexibility.

**Study on the number of query points.** We ablate with different numbers of query points $N_s$ in Tab. 2d. In general, the performance is in proportion to the point quantity, although the increase of $N_s$ only benefits the performance a little. $N_s$ reaches the sweet spot at 21, where the performance no longer increases with larger $N_s$. To balance the model complexity and performance, we set $N_s$ to 21.

**Study on point-level deformation.** In Tab. 2e, we compare point-level deformable convolution and direct interpolation for point-level feature extraction. Using deformable operator achieves higher performance than direct interpolation at query point location, demonstrating that it is beneficial to consider local context with dynamic modeling in action detection.

**Study on instance-level mixing.** The effect of frame-wise and channel-wise mixing is studied in Tab. 2f. We first ablate with the single application of each mixing, i.e. frame-only or channel-only. Results show that the performance degrades more without frame mixing, demonstrating the importance of frame mixing over channel mixing in multi-label TAD. Furthermore, we explore different combinations of frame and channel mixing with two cascaded alternatives, i.e. frame mixing $\rightarrow$ channel mixing and channel mixing $\rightarrow$ frame mixing. Tab. 2f shows that compared with frame-only and channel-only performances respectively, cascaded designs backfire at the performance with a decrease of 1% avg-mAP at frame $\rightarrow$ channel and 3% avg-mAP at channel $\rightarrow$ frame. To tackle action detection in video domain, intra-proposal channel mixing does not help with subsequent frame mixing and vice versa.

**Study on the result fusion parameter $\beta$.** Combining sparse detection results with dense segmentation scores provides smoother frame-level scores for segmentation-mAP. We ablate with choices of $\beta$ on both datasets in Tab. 3a. $\beta$ is set to 0.2 for MultiTHUMOS and 0.96 for Charades based on empirical results.

**Study on the offset scaling parameter $s$.** This scaling parameter is conventional in box-based object detectors [32, 1, 35], which is to scale regression offsets with respect to the box size instead of the image size. We extend this design to our PointTAD. In Tab. 3b, we compare the regression offsets predicted with respect to action duration (a.k.a offset scaled by duration) and with respect to clip duration (offset without scaling) on MultiTHUMOS. The result demonstrates the effectiveness of this scaling strategy on point-based detectors.

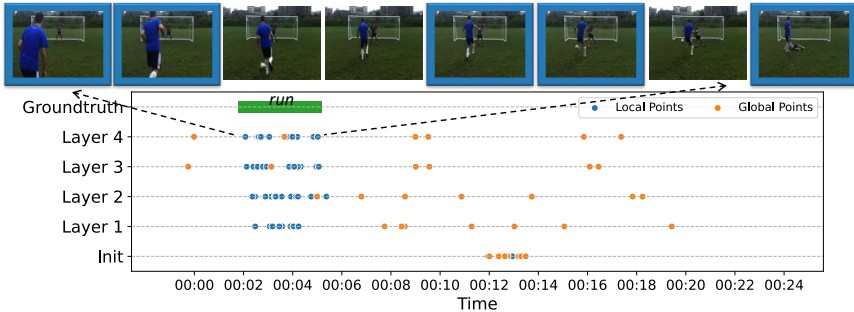

Figure 5: Visualization of **learned query points** and the corresponding action groundtruth on a sample from MultiTHUMOS.

## 4.4 Query Points Visualization

Fig. 5 illustrates the learned query points and the corresponding action target on a sample from MultiTHUMOS. Partial mix-max transformation divides the query points into confined local points (blue) and open-ended global points (orange). Local points are learned to attend action boundaries with two extreme points and semantic key frames with interior points. Global points sparsely distribute over the entire video for global context. Specifically, we observe that the interior local points effectively capture the most distinguished characteristic of target action *"Run"*, by capturing rapid forward movement of *both* legs and neglecting frames of similar yet disruptive movements: hopping and scoring. Moreover, the iterative refining process of query points are depicted from initial positions to the final positions in the last decoder layer. Query points are learned to kick off from the median position in the window. After the first decoder layer, the query points are already able to coarsely locate the action target. Then, these query points automatically converge to finer localization along the decoder layers.

## 5 Limitations and Future Work

PointTAD is proposed to solve the complex problem of multi-label TAD, by leveraging learnable query points for flexible and distinct action representation. Currently, we validate our model on two popular multi-label TAD benchmarks which include sport events and daily indoor activities. PointTAD achieves superior performance to all previous multi-label TAD methods as well as the state-of-the-art single-label TAD methods under detection-mAP metric. We have not demonstrated our model's ability in general action detection in more diverse scenarios, including important tasks such as action spotting, sentence grounding, and so on. In the future, we would continue to explore the advantages of point-based action representation in broader scope for video understanding. Meanwhile, our PointTAD training still relies on the intermediate supervision and we hope to design more effective training paradigm for the query-based detection pipeline.

## 6 Conclusion

In this paper, we have studied the complex multi-label TAD that aims to detect all actions from a multi-label untrimmed video. We formulate this problem as a sparse detection task and extend the traditional query-based action detection framework from single-label TAD. Faced with the challenge of concurrent instances and complex action relations in multi-label TAD, we introduce a set of learnable query points to effectively capture action boundaries and characterize action semantics for fine-grained action modeling Moreover, to facilitate the decoding process, we propose the Multi-level Interactive Module that integrates action semantics at both point level and instance level, by using dynamic kernels based on query vector. Finally, PointTAD yields an end-to-end trainable architecture by using only RGB inputs for easy deployment in practice. Our PointTAD surpasses all previous methods by a large margin under the detection-mAP and achieves promising results under the segmentation-mAP on two popular multi-label TAD benchmarks.

## Acknowledgments and Disclosure of Funding

This work is supported by National Natural Science Foundation of China (No. 62076119, No. 61921006), the Fundamental Research Funds for the Central Universities (No. 020214380091), and Collaborative Innovation Center of Novel Software Technology and Industrialization.

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
