# Supplementary Material for
# PointTAD: Multi-Label Temporal Action Detection with Learnable Query Points

**Jing Tan** [1*]   **Xiaotong Zhao** [2]   **Xintian Shi** [2]   **Bin Kang** [2]   **Limin Wang** [1,3†]

[1]State Key Laboratory for Novel Software Technology, Nanjing University
[2]Platform and Content Group (PCG), Tencent      [3]Shanghai AI Lab

jtan@smail.nju.edu.cn, {davidxtzhao,tinaxtshi,binkang}@tencent.com, lmwang@nju.edu.cn

## A  Appendix

### A.1  Comparison with the State-of-the-Art Single-label TAD Methods

In this sub-section, we report the detection-mAP results of some state-of-the-art single-label TAD methods (i.e. P-GCN [10], AFSD [1], ContextLoc [11]) reproduced on multi-label TAD task. Since MultiTHUMOS share similar data preparation with THUMOS14, the reproductions are conducted on MultiTHUMOS to get the best performance out of these models without additional hyper-parameter tuning. P-GCN [10] and ContextLoc [11] are both two-staged action detector that take in proposal generation results (e.g. BSN [2] proposals) and perform relation modeling based on coarse proposals to refine and classify each action candidate. AFSD [1] follows FCOS [8] to employ anchor-free architecture for TAD task. It is able to generate action detection results directly from the network, without pre-defined anchors or external video-level class labels.

The results in Tab. A show that direct application of single-label TAD models on multi-label TAD is deficient to achieve good detection performance, hence it is non-trivial to extend action detectors to multi-label TAD. Our PointTAD with strong image augmentation surpasses all of these state-of-the-art single-label TAD methods by a large margin, demonstrating the advance of our model to deal with concurrent instances and complex action relations as an action detector.

Table A: **Comparison with the state-of-the-art single-label TAD models** on MultiTHUMOS test set with under detection-mAP (%). The single-label TAD methods are reproduced with RGB input only.

| Methods | 0.1 | 0.2 | 0.3 | 0.4 | 0.5 | Average |
|---|---|---|---|---|---|---|
| BSN [2]+P-GCN [10] | 22.2 | 20.0 | 16.7 | 12.5 | 8.5 | 10.0 |
| BSN [2]+ContextLoc [11] | 22.9 | 21.0 | 18.0 | 14.6 | 10.8 | 11.0 |
| AFSD [1] | 30.5 | 27.4 | 23.7 | 19.0 | 14.0 | 14.7 |
| Ours | **42.3** | **39.7** | **35.8** | **30.9** | **24.9** | **23.5** |

### A.2  Error Bar

We follow the practice in [4] to compute the error bar for our model over 3 runs, and report the mean and standard derivation under detection-mAP in Tab. B. The implementation with stronger image augmentation appears to have larger derivation on MultiTHUMOS than the `Center_Crop` implementation.

---

[*]Work is done during internship at Tencent PCG.   [†]Corresponding author.

36th Conference on Neural Information Processing Systems (NeurIPS 2022).

Table B: **Error Bar** on MultiTHUMOS test set and Charades test set under detection-mAP (%).

| | Dataset | 0.1 | 0.2 | 0.3 | 0.4 | 0.5 | Average |
|---|---|---|---|---|---|---|---|
| PointTAD§ | MultiTHUMOS | 41.16±1.08 | 38.57±1.04 | 34.96±0.80 | 30.34±0.61 | 24.85±0.30 | 23.10±0.34 |
| | Charades | 17.71±0.41 | 17.05±0.38 | 15.98±0.39 | 14.74±0.28 | 13.20±0.25 | 11.92±0.23 |
| PointTAD | MultiTHUMOS | 39.20±0.10 | 36.66±0.04 | 33.32±0.03 | 28.51±0.07 | 22.90±0.36 | 21.51±0.06 |
| | Charades | 16.72±0.26 | 16.11±0.28 | 15.17±0.26 | 14.00±0.26 | 12.68±0.29 | 11.36±0.20 |

§ *indicates results trained with stronger image augmentation as in [3].*

## A.3 Evaluation on THUMOS14

Following RTD (query-based TAD method), we use the same feature representation and place our PointTAD head on top to build a direct TAD detector. Note that our TAD detector does not reply on the video-level classifier for action recognition and directly produce the action labels with our own PointTAD head. The result on the THUMOS14 dataset is reported in the Tab. C. We obtain better performance on this single-label TAD dataset, demonstrating the generalization ability of PointTAD to various TAD datasets.

Table C: **Evaluation on standard TAD benchmark THUMOS14** under detection-mAP (%).

| Methods | 0.3 | 0.4 | 0.5 | 0.6 | 0.7 | Average |
|---|---|---|---|---|---|---|
| RTD + UNet | 58.5 | 53.1 | 45.1 | 36.4 | 25.0 | 43.6 |
| PointTAD | 62.6 | 55.9 | 46.2 | 35.3 | 22.8 | 44.6 |

## A.4 Comparison with Query-based Baselines

In the ablation study of the main paper, we have shown the comparison between PointTAD and a Sparse-RCNN based baseline (segment-based variant), which proves the effectiveness of point representation. We have implemented another DETR based baseline on the MultiTHUMOS dataset. The performance comparison is reported in Tab. D, and our PointTAD obtains better results thanks to our more flexible point-based representation.

Table D: **Comparison with query-based baseline** under detection-mAP (%).

| Methods | 0.2 | 0.5 | 0.7 | Avg |
|---|---|---|---|---|
| DETR-alike baseline | 26.1 | 16.9 | 7.7 | 15.5 |
| Sparse R-CNN alike baseline | 33.1 | 20.1 | 9.8 | 19.4 |
| PointTAD | **36.6** | **22.8** | **10.6** | **21.5** |

## A.5 Other Training Details

In this sub-section, we share some of our observations in building an end-to-end trainable architecture for multi-label TAD via ablations on input image resolution and the number of decoder layers $L$.

**Study on input image resolution.** In Tab. Ea, we show the detection performance with different input image resolution. According to [3], cropping images to $128^2$ is adequate to tackle single-label TAD. However, we observe from the experiments that handling multi-label TAD requires more spatial information to distinguish concurrent instances from different categories and $128^2$ image resolution is far less sufficient to solve the task. Our detection performance improves greatly by the increase of image resolution: from $96^2$ to $128^2$ the detection-mAP improves by absolute 3.9% at average det-mAP, from $128^2$ to $160^2$ the performance improves by absolute 1% at average det-mAP. The performance gain slows down at larger image resolution: 0.6% gain at average det-mAP from $160^2$ to $192^2$. We settled at $192^2$ to balance the trade-off between memory consumption and model performance.

Table E: Ablation experiments w.r.t. **E2E training** on MultiTHUMOS. Default setting is colored gray .

(a) **Input spatial resolution**: from $96^2$ to $192^2$.

| $H \times W$ | 0.2 | 0.5 | 0.7 | Avg |
|---|---|---|---|---|
| $96 \times 96$ | 28.1 | 16.4 | 7.5 | 16.0 |
| $128 \times 128$ | 33.8 | 21.3 | 10.1 | 19.9 |
| $160 \times 160$ | 35.8 | 22.4 | 10.2 | 20.9 |
| $192 \times 192$ | **36.6** | **22.8** | **10.6** | **21.5** |

(b) **Number of decoder layers** $L$.

| $L$ | 0.2 | 0.5 | 0.7 | Avg |
|---|---|---|---|---|
| 3 | 35.1 | 21.7 | 9.0 | 20.2 |
| 4 | **36.6** | **22.8** | **10.6** | **21.5** |
| 5 | 35.9 | 22.6 | 10.2 | 21.1 |
| 6 | Out of Memory | | | |

**Study on the number of decoder layers.** The choice of $L$ is influenced by end-to-end training due to memory consumption of video encoder. We carefully decrease $L$ from the common setting of $L = 6$, as shown in Tab. Eb. Results indicate that 4-layer and 5-layer designs are quite similar in performance, yet from 4-layer design down, the decrease of $L$ leads to obvious performance degrade. Hence, $L$ is set to 4 empirically.

## A.6 Visualization

We show detailed visualizations of learnable query points of PointTAD in Fig. A. The visualizations are conducted on the samples with concurrent actions and multiple action categories from the test set of MultiTHUMOS (Fig. Aa) and Charades (Fig. Ab), covering different video scenarios such as daily events and sports matches. The first row briefly visualizes RGB frames of the video. The second row plots the temporal locations of groundtruth actions and the query points from the query that best predicts the action. In the rest of the figure, the left column shows action frames inside the groundtruth, where the semantic keyframes decided by the local query points are highlighted in blue. In the right column are selected frames corresponding to global query points. From the figure, we can see that the local query points not only learn different sets of representations for concurrent actions, but also capture important frames that indicate the action semantics. Global query points tend to distribute uniformly in the video clip and capture mostly close-up or background frames for sport events, possibly in the purpose of providing supporting background information for temporal action detection.

## A.7 Societal Impacts

This paper proposes PointTAD, a solution with learnable query points to tackle multi-label TAD. PointTAD is the first to introduce points/keyframes for segment-level video representation. Such practice addresses the non-uniform temporal structure of videos well and could potentially drive the development of general video understanding systems for finer point-based representation. The potential applications include video editing, anomaly event detection, etc. PointTAD enables end-to-end inference with raw video input, which benefits the deployment of automated online services for batch video processing, saving lots of human effort from offline, video-per-video handling. As the model is data-driven, any bias in training data could be captured in the algorithm. Apart from this aspect, there are no known ethical issues in the real-world applications of this technology.

## A.8 Code and License

Our codebase is mainly built upon RTD-Net[2] [7] protected by Apache-2.0 License and Sparse R-CNN[3] [6] protected by MIT license. MultiTHUMOS [9] dataset and Charades [5] dataset are restricted to non-commercial use only.

---

[2]https://github.com/MCG-NJU/RTD-Action
[3]https://github.com/PeizeSun/SparseR-CNN

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

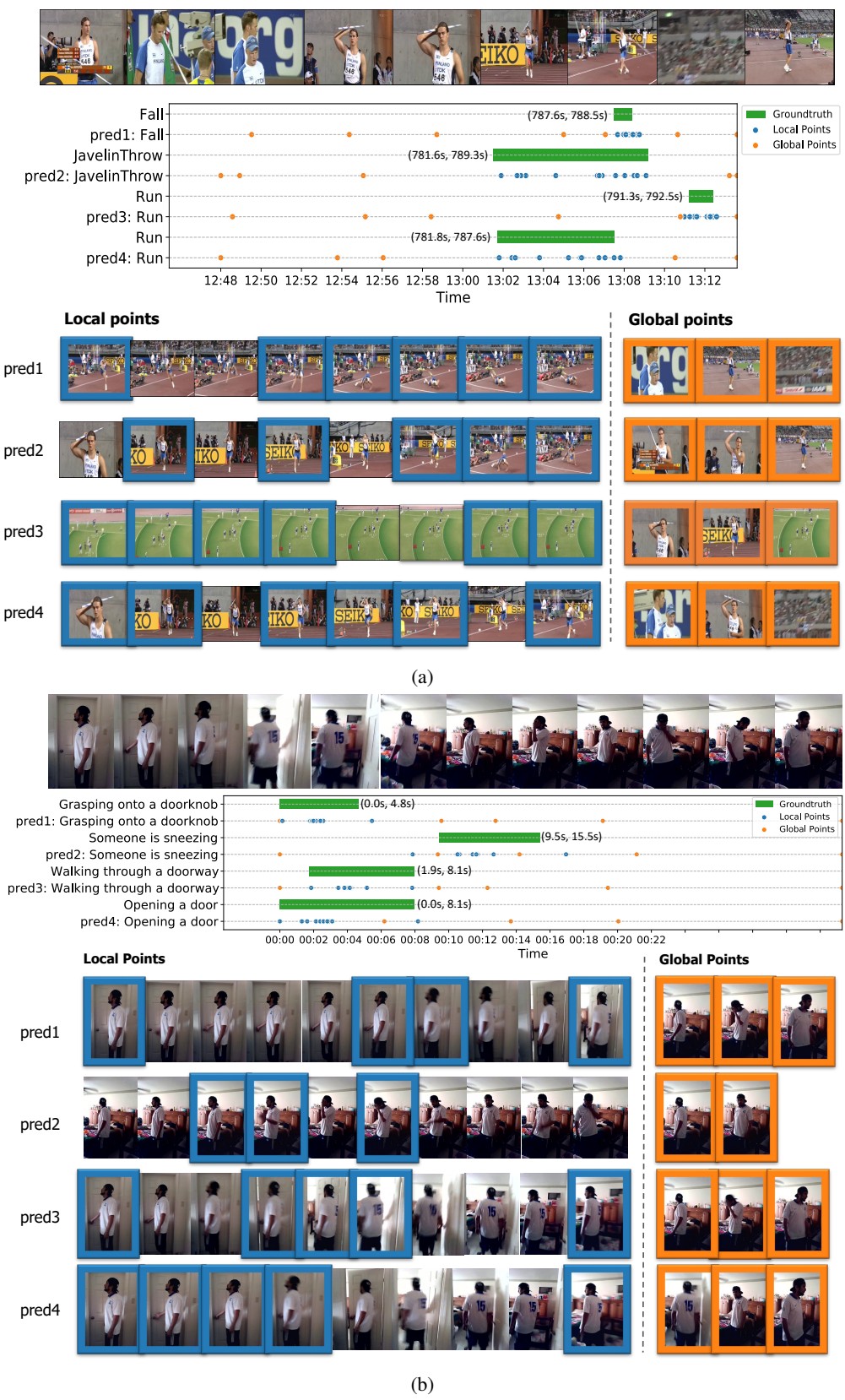

Figure A: Visualizations of the learnable query points of PointTAD on (a) MultiTHUMOS and (b) Charades.