# OpenReview forum: "PointTAD: Multi-Label Temporal Action Detection with Learnable Query Points"
_NeurIPS.cc/2022/Conference — NeurIPS 2022 Accept_

### Official Review · Reviewer_eH4Z · 2022-07-11

**Rating:** 8
**Confidence:** 4
**Soundness:** 4 excellent
**Presentation:** 4 excellent
**Contribution:** 4 excellent

**Summary:**

This paper identifies the impractical setup of classic single-label TAD and solves a more complex problem of multi-label TAD. It presents a novel query-based action detector with action point representation and multi-level interactive module to handle the co-occurring actions and fine-grained discrimination between categories. Extensive experiments on MultiTHUMOS and Charades demonstrate the effectiveness of the proposed method under detection metrics.

**Questions:**

Please refer to the 'Weaknesses' for the detailed questions.

**Limitations:**

Yes, discussed in supplements.

**Strengths And Weaknesses:**

Strengths:
+ This paper is the first to discuss the reason behind wide usage of video-level classifiers in traditional TAD, which attributes to the label-deficient traditional TAD benchmarks. It provides the first instance detection baseline for the more complex multi-label TAD with detection metrics and is very likely to encourage future TAD works to validate on these challenging multi-label benchmarks.
+ The proposed method tackles concurrent instances and fine-grained classification by introducing query points to replace action segments, for flexible capture of boundary and semantic information at the same time. To my knowledge, PointTAD is the first to integrate point detector with query-based detector in the field of temporal action detection.
+ The proposed method improves action decoding by designing a comprehensive local-to-global temporal modeling module at pointwise, intra-proposal (Instance-level) and inter-proposal (MHSA).
+ Experimental results under detection-mAP are good and surpass previous single-label and multi-label methods.
+ Ablations are solid. The performance improvement by newly designed modules, i.e., query points and multi-level interactive module are verified in ablations. Other important parameter choices are also discussed in ablations.
+ Detailed visualizations (fig. 5, fig. A) demonstrate the ability of query points to capture different essential motion cues for fine-grained actions.
+ The paper is well written and easy to follow, with motivation highlighted and major contributions well organized. The difference of this work and other query-based detectors is clearly addressed in the related work.

Weaknesses：
- The segmentation-mAP is weaker than previous SOTA. However, it’s understandable as instance detectors commonly get outperformed by segmentation methods under segmentation metrics. I would suggest the authors to try using sparse detections to refine SOTA segmentation results to improve segmentation-mAP (could use the practice in [M.1] but reversed).
- Do the Nq action queries share the same set of query points, or each query has its own query points?
- Qualitative results are suggested to add the comparison with segment-based variant to show the improvement of query points over action segments.
- The wording in Line 42 is inaccurate. Segment-based detectors can at least detect some of the ground truth actions, it should be “segment-based action detectors mainly predict two kinds of error predictions”.
- The captions of S1 and S2 in fig1 should specify the basic unit for the timestamps, e.g., second.

[M.1] "Cdc: Convolutional-de-convolutional networks for precise temporal action localization in untrimmed videos.", Zheng et al., ICCV 2017.

---

> ### Author Response · Authors · 2022-08-02
> **Responses to Reviewer eH4Z**
>
> We thank the reviewer for the positive and detailed feedback. Our response is summarized as follows:
>
> **Q.1** *Refine SOTA segmentation results with PointTAD sparse detections.*
> **R.1** The segmentation-mAP of fusing PointTAD predictions with MS-TCT segmentation results based on Eq (13) is 46.9\% for MultiTHUMOS and 26.8\% for Charades.
>
> **Q.2** *Do the $N_q$ action queries share the same set of query points?*
> **R.2** The learned embedding of query points are shared across samples, but different within the $N_q$ action queries.
>
> **Q.3** *Qualitative comparison with segment-based variant is suggested.*
> **R.3** We added the qualitative comparison with segment-based baseline to Fig. 4 of the revised paper, please check it out.
>
> **Q.4** *The wording and captions in introduction needs revision.*
> **R.4** We have revised the argument as well as the figure, please check the updated paper.

---

### Official Review · Reviewer_QSXY · 2022-07-11

**Rating:** 7
**Confidence:** 5
**Soundness:** 3 good
**Presentation:** 3 good
**Contribution:** 3 good

**Summary:**

The paper proposes a learnable query points-based method for multi-label temporal action detection, a more challenging and realistic task compared to single-level temporal action detection, aka. temporal action localization. Instead of uniform sampling on pre-defined segments, PointTAD uses a set of learnable query points to indicate the frames to attend to for each segments. Further, it applies frame mixing and channel mixing on the segment-level feature to integrate instance-level semantics.  The entire model takes RGB frames only as input and achieves significant improvement on detection-level mAP on MultiTHUMOS and Charades datasets.

**Questions:**

1. In L153-154, "*partial Min-Max* function is to select a subset of query points". How is the subset of local points determined?


==== Post-rebuttal Revision
My questions have been well addressed by the authors. My previous concerns are mostly on the presentation and I've lifted the quality of the presentation from "2 fair" to "3 good" and the overall rating from "6 weak accept" to "7 accept".

**Limitations:**

The authors haven't discussed the limitations and potential negative societal impact of their work.
Some potential one might include:
+ High computation cost of training video models end-to-end.
+ Potential biases in existing video datasets.

**Strengths And Weaknesses:**

*** Strengths

+ Using learnable query points to select representative frames for segment-level video representation is a novel idea, and proves to be more effective than uniform sampling +  Temporal RoIAlign (Table 2a). This might potentially bring some insights to the broader community of general video understanding since the video clips/segments are non-uniform by nature, especially in untrimmed videos.

+ The method achieves significant improvement on detection-level mAP on two datasets. Particularly, the detection mAP is more than doubled on Charades.

+ The paper conducts extensive ablations on the design choices. Most of improvements can be clearly explained by (1) query point-based representation v.s. segment-based representation and (2) the parallel application of frame mixing and channel mixing.


*** Weaknesses

- Some technical details are unclear. Some questions may appear in the next section.
  - How doe "partial Min-Max" select the subset query points?
  - What's the choice of the query numbers?
  - Eq (9) and Eq (10) look similar. Do you first do label assignment (Eq. 9) off the shelf and then optimize the network based on Eq. 10, or do you do them simultaneously?

- In L271-272, the authors argue that "*NoHuman* class is not a well-defined action category that has paired action boundaries". You can also report the detection-mAP by excluding NoHuman class to show it quantitatively.

- From the introduction, "the action decoder accomplishes context modeling at **point-wise**, **intra-proposal** and **inter-proposal** levels". How is the inter-proposal level modeled? The Multi-level Interactive Module is operated within individual proposal, isn't it? Any further operation on top?

---

> ### Author Response · Authors · 2022-08-02
> **Responses to Reviewer QSXY**
>
> Thank you for your positive feedback and constructive suggestions regarding our work. We address the comments as below:
>
> **Q.1** *How does "partial Min-Max" select the subset query points?*
> **R.1** Sorry for the confusion. For each query, we randomly take $\frac{2}{3}N_s$ query points from the point set of size $N_s$ to form the $\mathcal{P}_{local}$.  We have added this detail in the revised paper in section 3.2.
>
> **Q.2** *What's the choice of the query numbers?*
> **R.2** Thanks for your comment. The query number $N_q$ is set to 48 for both benchmarks. We have added it to section 4.1 in the revised paper.
>
> **Q.3** *Eq (9) and Eq (10) look similar. Do you first do label assignment (Eq. 9) off the shelf and then optimize the network based on Eq. 10, or do you do them simultaneously?*
> **R.3** Sorry for the confusion. We made a typo in Eq (10): the $\sigma(n)$ should be $\sigma_*(n)$, as it indicates the desired permutation calculated by Eq (9). The label assignment is done off the shelf at each iteration and the network is then optimized based on Eq (10). We have corrected this in the revised paper.
>
> **Q.4** *Detection-mAP without NoHuman class is suggested.*
> **R.4** Thanks for the advice. According to your suggestion, we add the results of detection-mAP without NoHuman class to Table 1 in the revised paper.
>
> **Q.5** *How is the inter-proposal level modeled?*
> **R.5** Thanks for your comment. Each action decoder includes the Multi-level Interactive Module and an MHSA for action queries (as illustrated in Fig.2 and Line 123-126 of the submission). Each query represents an action proposal and so the inter-proposal modeling is conducted via the MHSA.

---

### Official Review · Reviewer_tdcx · 2022-07-12

**Rating:** 5
**Confidence:** 5
**Soundness:** 4 excellent
**Presentation:** 3 good
**Contribution:** 3 good

**Summary:**


This paper introduces PointTAD, an architecture for temporally detecting activities in videos that contain multiple co-occurring activities of different labels.
The key idea is to use a sequence of query vectors, where each vector aims to predict an activity instance (or background) (similar to DETR). A key contribution of this work is that it associates a set of learnable points to each query vector, which points try to cover the duration of the activity instance and are iteratively refined. It also models intra-proposal relationships among the query-points as well as inter-instance relationships among the query vectors. The method is evaluated on MultiTHUMOS and Charades, where it leads to improved performance under the newly proposed detection-mAP metric, and competitive performance under the classic per frame mAP metric.



**Questions:**

Questions for rebuttal
==========
1. How was the threshold chosen for the post-processing of other methods’ results in Table 1?
2. Ideally it would be good to evaluate the method on Thumos or ActivityNet or explain why this is not possible.
3. Adding ablations for $beta$ and $s$, as well as a DETR-based baseline would strengthen the paper.

Suggestions
1. What is the motivation of 4 deformable sub-points? (instead of 2 etc)
2. State-of-the-art table is missing a lot of methods, I would suggest taking a look at the tables from [Nagwal, Activity Graph Transformer for Temporal Action Localization, arxiv21] and [Dai, CTRN: Class Temporal Relational Network for Action Detection, BMVC 2021].
3. How did you obtain the segments in the ablation study (Table 2a)? Is it from the query points (partial min-max)?
4. How did you choose the hyperparameters, e.g., for the loss weighting?


**Limitations:**

Yes

**Strengths And Weaknesses:**

Strengths
===========
1. This paper addresses an important problem, i.e. temporally detecting activities in videos that contain multiple co-occurring activities of different label. It also introduces a metric for evaluating the segment-level prediction of activity instances (instead of frame-based).
2. Ablations show that the proposed deformable convolution and the mixing strategy improve performance.
3. The method leads to improved detection-mAP on Charades and MultiThumos.
4. The idea of using query points alongside the query vectors is interesting and seems to be working nicely for pooling features to describe temporal segments.

Weaknesses
===========
1. Unfair comparison with existing per-frame labeling approaches with the detection-mAP metric: Although the addition of the detection-mAP metric is an important contribution of this work, comparison with prior work under this metric seems not to be fair for the compared approaches, since they do not directly predict segments. This paper post-processes the per-frame action predictions of these works based on thresholding in order to generate segments (from consecutive frames with predictions above a threshold). However, this is a very naive post-processing approach, which is also very sensitive to the choice of threshold (other options would be to detect peaks/blobs in the score time-series, which also involve hyperparams). Therefore, the big improvements of the proposed approach under this metric could be because of the sub-optimal choice of post-processing of the results of competing methods. Under the segmentation-mAP metric, the proposed approach is lagging behind SOTA. (Also, state-of-the-art numbers are missing. Even if other methods use optical flow, results should be reported and discussed).
2. Missing baseline: it seems like a baseline would be to use just query vectors, the Multi-Head Self-Attention head and predict proposals and class. This would be very similar to DETR and improvements over it would motivate the need for query points/for the multi-level interactive module.
3. Missing important ablations: What is the benefit of combining sparse predictions with dense scores instead of using just the dense scores (\beta=0)? What is the importance of the scaling parameter $s$ which differentiates the current approach from RepPoints[38]?
4. Choice of datasets: ActivityNet and THUMOS indeed don’t have many co-occurring activities like Charades/MultiThumos, but they are the standard benchmarks for activity detection methods that predict instances (with start/end times) instead of per-frame activity predictions. Evaluating on either one of them would allow comparison with stronger methods for detection (instead of applying a naive post-processing step on frame labeling methods as done now on Charades/MultiThumos).
5. Relation with existing modules is not explicitly described: The query points module adapts RepPoints[8] to the activity detection task,  the Point-level Locality Preservation uses ideas from deformable DETR, etc.
6. Some equations/statements need clarification: For example, a) how does the query vector predict $N_s$ offsets (ln 140?), b) $\mathbf{q} is defined to be NqxD in ln 119, but in later equations it seems to refer to one of the N_q vectors., c) The Linear() functions in eq 3, 4, 6, 7 could be explained in more detail.

---

> ### Author Response · Authors · 2022-08-02
> **Responses to Reviewer tdcx (Part 1/2)**
>
> We thank the reviewer for the detailed comments and constructive suggestions for improvement. Our response to the reviewer's comments is as below.
>
>
> ### Questions for Rebuttal
> **Q.1** *How was the threshold chosen for the post-processing of frame-based methods’ results in Table 1?*
> **R.1** Thanks for your comment. We set the threshold to 0.5 to produce binary predictions according to MLAD[32] and MS-TCT[7]. As the reviewer points out this post-processing could be sensitive to thresholds, we also experiment with different thresholds on the previous detection-mAP SOTA PDAN [8] during rebuttal. It turns out the threshold indeed affects the performance. We have updated the detection-mAP of frame-based methods under the optimal threshold in the revised paper. Nevertheless, PointTAD still surpasses the best detection-mAP among all thresholds (MultiTHUMOS: **21.5** vs 17.3; Charades: **11.1** vs 8.5), consistently demonstrating the effectiveness of our model.
>
> | Threshold   | 0.1  | 0.3  | 0.5  | 0.7  | 0.9  |
> |-------------|------|------|------|------|------|
> | MultiTHUMOS | 15.6 | 17.3 | 17.1 | 15.3 | 11.8 |
> | Charades    | 8.5  | 6.8  | 5.0  | 3.2  | 1.4  |
>
> **Q.2** *Evaluation on single-label TAD datasets.*
> **R.2** Thanks for your comment. Following RTD (query-based TAD method), we use the same feature representation and place our PointTAD head on top to build a direct TAD detector. Note that our TAD detector does not reply on the video-level classifier for action recognition and directly produce the action labels with our own PointTAD head. The result on the THUMOS14 dataset is reported in the below Table. We obtain better performance on this single-label TAD dataset, demonstrating the generalization ability of PointTAD to various TAD datasets.
> | THUMOS14   | 0.3  | 0.4  | 0.5  | 0.6  | 0.7  | avg  |
> |------------|------|------|------|------|------|------|
> | RTD + UNet | 58.5 | 53.1 | 45.1 | 36.4 | 25.0 | 43.6 |
> | PointTAD   | 62.6 | 55.9 | 46.2 | 35.3 | 22.8 | 44.6 |
>
> **Q.3** *Ablations for fusion parameter $\beta$ and scaling parameter $s$.*
> **R.3** 1) $\beta$: Combining sparse detection results with dense segmentation scores provides smoother frame-level scores for segmentation-mAP. We ablate with choices of $\beta$ on both datasets in the table below. $\beta$ is set to 0.2 for MultiTHUMOS and 0.96 for Charades based on empirical results.
>
> | $\beta$     | 0    | 0.2      | 0.4  | 0.6  | 0.8  | 0.96     | 1    |
> |-------------|------|----------|------|------|------|----------|------|
> | MultiTHUMOS | 33.0 | **39.8** | 39.2 | 38.1 | 37.3 | 36.8     | 35.9 |
> | Charades    | 13.8 | 14.3     | 15.1 | 16.6 | 19.2 | **21.0** | 18.7 |
>
>
> 2)$s$: This scaling parameter is conventional in box object detectors (Faster R-CNN, Cascade R-CNN and Sparse R-CNN[30]), which is to scale regression offsets with respect to the box size instead of the image size. We extend this design to our point detector. In the table below, we compare the regression offsets predicted with respect to action duration (offset scaled by duration) and with respect to window size (offset without scaling) on MultiTHUMOS.
>
> | $s$                     | 0.1  | 0.2  | 0.3  | Avg  |
> |-----------------------|------|------|------|------|
> | scale window size     | 38.8 | 36.4 | 32.6 | 21.0 |
> | scale action duration | 39.1 | 36.6 | 33.0 | 21.5 |
>
>
> **Q.4** *Comparison with DETR-based baseline.*
> **R.4** Thanks for your suggestion. In Table 2a of submission, we have shown the comparison between PointTAD and a Sparse-RCNN based baseline (segment-based variant), which proves the effectiveness of point representation. According to your suggestion, we have implemented another DETR based baseline on the MultiTHUMOS dataset. The performance comparison is reported in the below table, and our PointTAD obtains better results thanks to our more flexible point-based representation.
> | Methods                     | 0.1  | 0.2  | 0.3  | Avg  |
> |-----------------------------|------|------|------|------|
> | DETR-alike baseline         | 28.2 | 26.1 | 23.6 | 15.5 |
> | Sparse R-CNN alike baseline | 35.4 | 33.1 | 30.0 | 19.4 |
> | PointTAD                    | 39.1 | 36.6 | 33.0 | 21.5 |

---

> ### Author Response · Authors · 2022-08-02
> **Responses to Reviewer tdcx (Part 2/2)**
>
> ### Other Weakness
> **Q.5** *Relation with RepPoints and deformable DETR is not explicitly described.*
> **R.5** We have added an independent subsection to related work to discuss our relations with point-based detectors, please check the revised paper. The relation to deformable DETR is also added in Line 56 of the revised paper.
>
>
> **Q.6** *Equations / Statements need clarification.*
> **R.6** a) The $N_s$ query point offsets are predicted by Linear layer (input dimension is D, output dimension is $N_s$) from query vectors. We have clarified this in section 3.2 of the revised paper.
>
> b) **q** refers to the $N_q$ query vectors. We have revised section 3.3 for notation consistency in the revised version, thanks for pointing it out :)
>
> c) The linear layer in Eq (3)(4)(6)(7) are all implemented with fully connected layers. The input and output dimension of linear layers in Eq (3)(4)(6)(7) are reported in the table below. We also clarified the input and output dimensions in the revised paper.
>
> | Linear     | Eq (3) | Eq (4) | Eq (6) | Eq (7) |
> |------------|--------|--------|--------|--------|
> | Input Dim  | D   | D    | D    | D    |
> | Output Dim | 4    | $N_s$  | D′  | D′   |
>
>
> ### Suggestions
> **Q.7** *Motivation behind the number of deformable sub-points.*
> **R.7**  We set this hyperparameter to 4 according to the number of sampling points in TadTR [21] for temporal deformable attention. We have tried 2 as the temporal coordinate has binary directions, but this setting achieves slightly weaker performance: avg-mAP = 21.4\% on MultiTHUMOS.
>
> **Q.8** *Missing some state-of-the-art methods in comparison table.*
> **R.8** We have added more methods to the comparison table and included methods with Optical Flow input, please check Table 1 in the revised paper.
>
> **Q.9** *How to obtain the segments in Table 2a? Is it from the query points (partial min-max)?*
> **R.9** The segments are NOT pseudo segments converted from query points. In fact, this segment-based baseline is similar to Sparse R-CNN (but with parallel mixing for ablation purposes), where actions are represented as segments by paired start-end positions.
>
> **Q.10** *How to choose the hyperparameters?*
> **R.10** We determine most of the hyper-parameters, such as loss weighting, by empirical results. Some parameters, such as the number of deformable sub-points, the input temporal resolution and input frames for each sample are decided based on the experience from previous works ([20], [21]).

---

> ### Author Response · Authors · 2022-08-09
> **Friendly reminder of less than 24 hours to discuss**
>
> Dear Reviewer tdcx:
>
> Thank you again for the constructive suggestions on our paper. There’s less than 24 hours till the discussion deadline, we would like to know if there’s any unresolved questions that we can help with. Please feel free to comment and we would try our best to address your concerns :)
>
> Have a nice day,
> Authors of Paper 1881

---

### Official Review · Reviewer_ZWpN · 2022-07-12

**Rating:** 4
**Confidence:** 4
**Soundness:** 3 good
**Presentation:** 2 fair
**Contribution:** 2 fair

**Summary:**

This paper focuses on the complex multi-label temporal action detection that aims to localize all action instances from a multi-label untrimmed video. Existing query-based action detectors employ a segment to represent an action instance, which is insufficient to handle the concurrent instances and their richer relations. To mitigate this issue, this paper introduces a small set of learnable query points to represent important frames of each action instance. PointTAD provides a flexible mechanism to localize the discriminative frames at boundaries and as well the important frames inside the action.

**Questions:**

See Weaknesses.

**Limitations:**

The author did not provide the limitations and potential negative societal impact of their work.

**Strengths And Weaknesses:**

Strengths: This paper is well written and easy to understand.

Weaknesses:
1. The idea of using points to represent keyframes or objects is not very new, as it was already mentioned in [1,2,3,4]. The authors should add a subsection to the related work section summarizing the current related work and explaining the differences from these approaches.
2. From Table 1, the method proposed in this paper does not have a significant performance improvement over the SOTA methods such as MS-TCT.
3. More related methods such as [1,2,3,4] should be compared and discussed in the experimental part.

[1]Guan, Genliang, Zhiyong Wang, Shiyang Lu, Jeremiah Da Deng, and David Dagan Feng. "Keypoint-based keyframe selection." IEEE Transactions on circuits and systems for video technology 23, no. 4 (2012): 729-734.
[2]Zhou, Xingyi, Vladlen Koltun, and Philipp Krähenbühl. "Tracking objects as points." In European Conference on Computer Vision, pp. 474-490. Springer, Cham, 2020.
[3]Tang, Hao, Hong Liu, Wei Xiao, and Nicu Sebe. "Fast and robust dynamic hand gesture recognition via key frames extraction and feature fusion." Neurocomputing 331 (2019): 424-433.
[4]Li, Yixuan, Zixu Wang, Limin Wang, and Gangshan Wu. "Actions as moving points." In European Conference on Computer Vision, pp. 68-84. Springer, Cham, 2020.

---

> ### Author Response · Authors · 2022-08-02
> **Responses to Reviewer ZWpN**
>
> We thank the reviewer for the feedback. Below is our response to the comment.
>
> **Q.1** *The idea of using points to represent keyframes or objects is not very new. Discussion of related research is suggested to add into related work.*
> **R.1** Thanks for your comment. Your mentioned works are different from our method in many aspects. Our PointTAD tackles multi-label temporal action detection by treating action as a set of temporal points (keyframes), while these mentioned papers all deal with different problems other than TAD and with different techniques.
> [1] uses local and spatial keypoints (SIFT) to extract frame-level features and proposes a greedy algorithm to choose keyframes. [3] selects keyframes based on low-level features and generate video feature for gesture recognition in a bottom-up manner. Instead, our PointTAD presents a top-down method to direct regress the temporal location of keyframes. [2] and [4] all use points to represent object tracks or action tracks, with a focus on representing the spatial location of objects or actions. We have added a subsection in related work to discuss our work with these point-based representations, please check our revised paper.
>
> > [1]Guan, Genliang, Zhiyong Wang, Shiyang Lu, Jeremiah Da Deng, and David Dagan Feng. "Keypoint-based keyframe selection." IEEE Transactions on circuits and systems for video technology 23, no. 4 (2012): 729-734.
> > [2]Zhou, Xingyi, Vladlen Koltun, and Philipp Krähenbühl. "Tracking objects as points." In European Conference on Computer Vision, pp. 474-490. Springer, Cham, 2020.
> > [3]Tang, Hao, Hong Liu, Wei Xiao, and Nicu Sebe. "Fast and robust dynamic hand gesture recognition via key frames extraction and feature fusion." Neurocomputing 331 (2019): 424-433.
> > [4]Li, Yixuan, Zixu Wang, Limin Wang, and Gangshan Wu. "Actions as moving points." In European Conference on Computer Vision, pp. 68-84. Springer, Cham, 2020.
>
>
>
> **Q.2** *No significant improvement over SOTA.*
> **R.2** Thanks for your comment. The main contribution of this paper is to focus on a new setting (multi-label TAD) in temporal action detection and also to introduce a new metric (detection-mAP) for this challenging setting. This contribution is acknowledged by the other reviewers. In Table 1, we show that PointTAD achieves the state-of-the-art performance under detection-mAP by large margin (also recognized by Reviewer tdcx, QSXY and eH4Z). In the rebuttal, we further add other experiments to illustrate the effectiveness of our PointTAD over DETR-alike baselines and on the standard TAD benchmark of THUMOS14.
>
>
> **Q.3** *Related methods such as [1,2,3,4] should be compared and discussed in experiments.*
> **R.3** Thanks for your comment. As we discussed in Q1, [1,2,3,4] tackle very different tasks from TAD and none of them generates temporal action proposals for evaluation. As much as we would love to, it is not feasible to directly compare these methods in experiments.

---

> > ### Comment · Reviewer_ZWpN · 2022-08-09
> > **Need more feedbacks**
> >
> > >Q.1 The idea of using points to represent keyframes or objects is not very new. Discussion of related research is suggested to add into related work.
> >
> > What is the difference between the method proposed in this paper and [2,4], which the author does not seem to mention?
> >
> > [2]Zhou, Xingyi, Vladlen Koltun, and Philipp Krähenbühl. "Tracking objects as points." In European Conference on Computer Vision, pp. 474-490. Springer, Cham, 2020.
> > [4]Li, Yixuan, Zixu Wang, Limin Wang, and Gangshan Wu. "Actions as moving points." In European Conference on Computer Vision, pp. 68-84. Springer, Cham, 2020.
> >
> > >Q.3 Related methods such as [1,2,3,4] should be compared and discussed in experiments.
> >
> > Although these two methods [2, 4] do not deal with the same task, they should be related to the method proposed in this paper, and in my opinion, they can be compared. Moreover, the author said in the newly added related work that the proposed method was improved based on RepPoints, but the author did not compare it with RepPoints in the experimental part including Table 1.
> >
> > ``Since the author did not address my concerns well, I keep the original score``

---

> > > ### Author Response · Authors · 2022-08-09
> > > **Responses to Reviewer ZWpN**
> > >
> > > > What is the difference between the method proposed in this paper and [2,4], which the author does not seem to mention?
> > >
> > > As we stated in Line 93 of the revised paper and in the first response,  [2] and [4] all use points to represent object tracks or spatiotemporal action tracks, with a focus on representing the **spatial** location of objects or actions and **do not adopt keyframes in temporal aspect**. In contract, PointTAD represents action as a set of **temporal** points (keyframes) and **does not directly interact with the spatial content of video frames** (the spatial resolution is compressed into 1024 channels by backbone network before PointTAD head).
> > >
> > > > Although these two methods [2, 4] do not deal with the same task, they should be related to the method proposed in this paper, and in my opinion, they can be compared. Moreover, the author said in the newly added related work that the proposed method was improved based on RepPoints, but the author did not compare it with RepPoints in the experimental part including Table 1.
> > >
> > > As stated above, [2] and [4] are very different from PointTAD for they use spatial centerpoints to represent objects/humans and we use temporal keyframes to represent actions.
> > >
> > > The reason that we cannot directly compare these methods with PointTAD is that : (a) [2] tackles tracking and [4] tackles spatiotemporal action detection, none of the two methods generates temporal action proposals; (b) [2] and [4] both require spatial bounding box supervision for training and TAD benchmarks do not have these annotations, therefore it's not feasible to even re-implement these models on TAD benchmarks.
> > >
> > > RepPoints tackles object detection, which is also a very different task from TAD. We follow the idea of representative point representation from RepPoints and adapt it in TAD to address the non-uniform temporal structure in videos. Sorry for the confusion from the wording of "improved over", we have revised this word choice in the paper.

---

> ### Author Response · Authors · 2022-08-09
> **Friendly reminder of less than 24 hours to discuss**
>
> Dear Reviewer ZWpN:
>
> Thank you again for your thoughtful feedback on our paper. There’s less than 24 hours till the discussion deadline, we would like to know if there’s any unresolved questions that we can help with. Please feel free to comment and we would try our best to address your concerns :)
>
> Have a nice day,
> Authors of Paper 1881

---

### Author Response · Authors · 2022-08-02
**General Response**

We sincerely appreciate all reviewers' efforts in reviewing our paper and giving insightful comments as well as valuable suggestions. We are glad to find that the reviewers generally acknowledge the following novelty and contributions of our work.
* **Framework.** Using learnable query points to select representative frames for instance-level action representation is novel [tdcx,QSXY,eH4Z] and is more effective over uniform sampling or temporal RoI Align [QSXY]. We hope our work will inspire general video understanding [QSXY] to opt for the more effective temporal action detection with point representation.
* **Experiments.** Experiments show improved detection-mAP performance on the two popular multi-label TAD benchmarks [tdcx,QSXY,eH4Z]. The improvement of query point representation, point-level locality preservation and instance-level parallel mixing are supported by ablations [tdcx,QSXY,eH4Z].

As suggested by the reviewers, we include the following contents in the revised manuscript to further strengthen our paper. The major revision is summarized as follows. Our detailed responses can be found in each response section to the reviewers.
* **Extended experiments** including evaluation on THUMOS14, ablation on result fusion parameter $\beta$ and offset scaling factor $s$, comparison to DETR-based baseline are added to the revised appendix [tdcx, QSXY].
* **Relation to point-based representation literature.** We have added an independent subsection in the revised related work to discuss our differences with keyframe selection literature and point-based detectors [ZWpN, tdcx].
* **Updates to comparison table.** In the revised paper, we have updated the detection-mAP of segmentation methods [tdcx], included more methods with optical flow input [tdcx] and added detection-mAP result without NoHuman class for MultiTHUMOS results [QSXY].
* **Clarifications on equations and statements.** We have clarified all the ambiguities mentioned by the reviewers in the revised manuscript [tdcx,QSXY,eH4Z].

---

### Author Response · Authors · 2022-08-07
**Have we addressed your concerns?**

Dear all reviewers:

Thanks for your suggestions on our paper.  As the reviewer-author discussion deadline is approaching, we would like to know whether our reply has addressed your concerns. If you have any questions, please feel free to let us know, and we will try our best to address your concerns.

Best,
Authors of Paper 1881

---

### Meta-Review · Area_Chair_rhJ9 · 2022-08-22

**Recommendation:** Accept
**Confidence:** Certain

**Metareview:**

This paper considers the problem of detecting temporal activities in videos which contain multiple co-occurring activities of different labels. It is an important problem that arises in many computer vision tasks. The paper is generally well written. Specifically, using learnable query points to select representative frames for segment-level video representation seems to be a novel idea. The experiment results also show promises of the proposed method. Nevertheless, a number of comments and questions were raised by the reviewers. We thank the authors for responding to them in detail and even revising their paper accordingly, which includes providing more experiment results to support their claims. The authors are recommended to further revise their paper by addressing the remaining comments raised.


**Award:**

No

---

### Decision · Program_Chairs · 2022-09-14

Accept